# MCA: Modality Composition Awareness for Robust Composed Multimodal Retrieval

## Abstract

Multimodal retrieval, which seeks to retrieve relevant content across modalities such as text or image, supports applications from AI search to contents production. Despite the success of separate-encoder approaches like CLIP align modality-specific embeddings with contrastive learning, recent multimodal large language models (MLLMs) enable a unified encoder that directly processes composed inputs. While flexible and advanced, we identify that unified encoders trained with conventional contrastive learning are prone to learn modality shortcut, leading to poor robustness under distribution shifts. We propose a modality composition awareness framework to mitigate this issue. Concretely, a preference loss enforces multimodal embeddings to outperform their unimodal counterparts, while a composition regularization objective aligns multimodal embeddings with prototypes composed from its unimodal parts. These objectives explicitly model structural relationships between the composed representation and its unimodal counterparts. Experiments on various benchmarks show gains in out-of-distribution retrieval, highlighting modality composition awareness as a effective principle for robust composed multimodal retrieval when utilizing MLLMs as the unified encoder.

## 1 Introduction

Multimodal retrieval, which aims to retrieve semantically relevant contents across multiple modalities such as text, image and audio, is a fundamental task in various information fields. Its applications span a wide range of domains, such as text-vision retrieval(Huynh et al., 2025; Wang et al., 2021), music retrieval(Doh et al., 2023), product search(Goenka et al., 2022; Zhu et al., 2024b), and multimodal retrieval-augmented generation (Yasunaga et al., 2023; Ghosh et al., 2024; Yang et al., 2024; Jeong et al., 2025). The core ability of multimodal retrieval is to represent multimodal inputs in a shared and comparable embedding space. A prevailing approach to this problem is to adopt unimodal encoders and align the encoded embeddings through contrastive learning (CL). Models following this separate-encoder paradigm, such as CLIP (Radford et al., 2021) and CLAP (Elizalde et al., 2023), have demonstrated the effectiveness of CL, achieving strong performance across various multimodal retrieval tasks. On the other hand, with the rapid development of multimodal large language models (MLLMs) (Alayrac et al., 2022; Li et al., 2023; Bai et al., 2023; Chu et al., 2023; Liu et al., 2023a; Chen et al., 2024), there has been growing interest in employing MLLMs as encoders for multimodal retrieval (Jiang et al., 2024; Zhang et al., 2024; Huang et al., 2025; Jiang et al., 2025). Unlike seperate-encoder frameworks, MLLMs are capable of processing inputs from different modalities, as well as their compositions, within a unified architecture, providing several advantages such as a shared semantic space across modalities, flexible handling of composed queries and documents (e.g., text+image or text+audio), and the ability to leverage powerful pretrained representations including rich language knowledge and multimodal understanding.

However, this flexibility also makes the model more prone to **modality shortcut learning**. Since all modalities are jointly processed within a shared architecture, the training loss can be minimized by over-relying on the stronger modality signal, while ignoring the complementary one. A toy example in Figure 1a illustrates that the model suffers from modality shortcuts when processing highly similar images, while ignoring the textual instruction: "put up convertible roof, remove snow, place SUV standing on flat asphalt". Illustration of learned representation with vanilla CL in Figure 1b also demonstrates that composed queries are poorly separated and often cluster near text-only queries, indicating that the model relies on modality shortcut instead of learning robust composed repre-

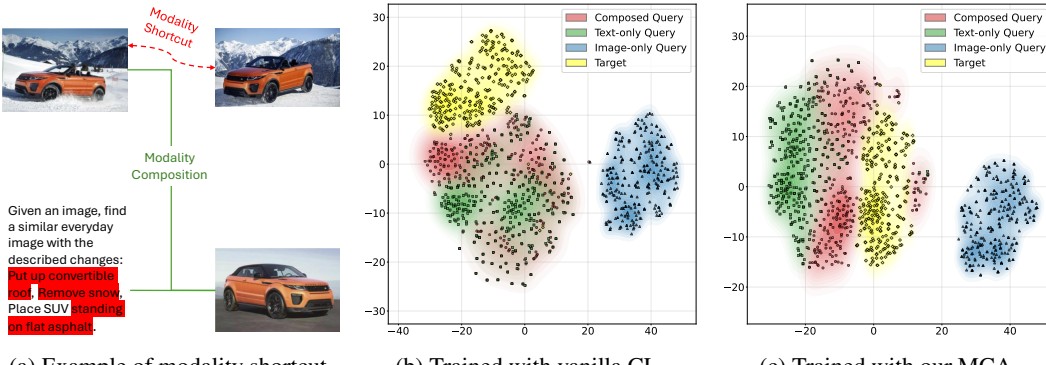

(a) Example of modality shortcut.     (b) Trained with vanilla CL.     (c) Trained with our MCA.

Figure 1: Illustration of modality shortcut problem in multimodal retrieval. **(a)** is a conceptual example where the red-highlighted text descriptions are ignored, leading the model to follow a shortcut path, instead of the expected composed representation. **(b)** and **(c)** are t-SNE visualizations of composed queries, unimodal queries and targets, that randomly sampled from CIRR dataset, under vanilla CL and our MCA, respectively. tSNE implementation details are introduced in §A.6.

sentations. Hence, the architecture shift from separate-encoder to unified-encoder makes the direct application of conventional CL objective to unified MLLM encoders limiting to robustness. Furthermore, the modality shortcut problem naturally extends to scenarios involving multiple modalities, particularly as recent work aims to unify various modalities into a single framework (Girdhar et al., 2023; Zhu et al., 2024a; Xu et al., 2025). As the number of modalities increases, so does the risk of the model collapsing to rely on a single dominant modality, making explicit modality composition-aware constraints crucial for robust generalization.

We argue that **modality composition** is the key to mitigate this issue. In this paper, we propose a *modality composition awareness* (MCA) framework that modeling the structural relationship between multimodal and unimodal representations, as shown in Figure 2. MCA consists of two complementary objectives from the preference and consistency perspective, respectively. First, a *preference loss* enforces that the embeddings of a multimodal composition should be more discriminative than any of its unimodal counterparts, thereby discouraging modality shortcut learning. Second, a *composition regularization* objective encourages the consistency between the composed embedding produced by the unified encoder and a compositional prototype constructed from its unimodel embeddings, ensuring that composed representations remain grounded in their constituent modalities. Together, these objectives explicitly model the structural relationship between multimodal and unimodal inputs, leading to more robust representation. Figure 1c visualizes the representation learned with MCA. It demonstrates that composed queries, unimodal queries and targets have a clearer boundary, showing that MCA reduces shortcut reliance.

To comprehensive evaluate MCA, we conduct extensive experiments on both in-domain (IND) and OOD benchmarks, covering retrieval and grounding tasks. The results demonstrate the MCA improves robustness with OOD improvements under distribution shifts while maintaining IND performance. Through ablations, we show that both the preference and regularization components contribute complementary benefits. In addition, we also highlight the interaction between input richness and the strength of our proposed losses. Those results suggest that explicitly modeling modality composition could be a general principle with broader implications for the unified multimodal retrieval using MLLMs.

## 2 RELATED WORK

**Multimodal Retrieval.** Classical multimodal retrieval typically adopts dual-encoder architectures, learning a text encoder and an image/audio/video encoder whose outputs are aligned in a share space via contrastive objectives. This line of work has achieved strong performance at scale and efficient retrieval (Radford et al., 2021; Li et al., 2022; Zhai et al., 2023). Beyond unimodal queries, composed retrieval focuses on queries that combine multiple modalities, such as text+image or text+audio, which better capture use intent in interactive and real-world scenarios. This setting

has been studied in contexts like text-guided image retrieval(Yu et al., 2016), fashion production search (Wu et al., 2021) and multimodal modification tasks (Liu et al., 2021b). Dual-encoder approaches are principally designed for unimodal queries. When the query or candidate is a composition of modalities, an extra fusion module (Wei et al., 2024; Huynh et al., 2025) has to be trained to fuse the modalities. While these methods demonstrate the feasibility of handling composed queries, they typically build upon conventional contrastive objectives and do not model the cross-modal interaction between the modalities, leading to a performance sacrifice (Huang et al., 2025)..

**MLLMs for Retrieval.** Recent MLLMs extend pretrained LLMs with multimodal adapters or cross-modal modules, enabling an integrated architecture to process and understand over text images, audios and videos (Alayrac et al., 2022; Li et al., 2023; Bai et al., 2023; Chu et al., 2023; Liu et al., 2023a; Chen et al., 2024). Given the unified processing of multiple modalities and rich pre-learned knowledge, MLLMs are increasingly applied to retrieval tasks, especially composed retrieval. Most approaches adapt the unified encoder directly with contrastive learning for embedding-based retrieval (Jiang et al., 2024; Zhang et al., 2024; Huang et al., 2025; Jiang et al., 2025), while others employ MLLMs as re-rankers for the later re-ranking (Lin et al., 2024). In this work, we study the embedding-based approaches. An advantage of MLLMs for retrieval is the capability of jointly understanding contents in multiple modalities with a unified encoder. However, adopting the unified encoder in contrastive learning could also lead to shortcut learning (Geirhos et al., 2020; Wu et al., 2022). As a result, these recent methods generally inherit conventional contrastive loss and therefore remain vulnerable to modality shortcut when handling composed inputs. Orthogonal to these previous works, we firstly study the modality shortcut problem in this setting and our proposed MCA framework mitigates issue by introducing modality composition-aware objectives that can be easily integrated with MLLM-based retriever to enhance the robustness.

## 3 METHODOLOGY

### 3.1 PRELIMINARY

Multimodal retrieval aims to search for target documents from a pool given a query. The embedding model, which is central to multimodal retrieval and the focus of our approach, is introduced in this section along with its encoding process and training objectives.

**Unified Multimodal Encoder.** Unlike conventional separate-encoder methods such as CLIP (Radford et al., 2021; Zhai et al., 2023), we study the approaches (Lin et al., 2024; Jiang et al., 2024; Zhang et al., 2024; Huang et al., 2025; Jiang et al., 2025) using a *unified encoder* $f_\theta(\cdot)$, e.g., a MLLM, in this paper. Formally, given an input $x$ that could comprises multiple modalities, $f_\theta(\cdot)$ encodes the input into a shared space as: $\mathbf{h} = f_\theta(x)$, where $\mathbf{h} \in \mathbb{R}^d$ is the embedding in $d$ dimensions. In addition, $f_\theta(\cdot)$ includes the necessary processes such as tokenization and pooling, and we omit those details because it does not affect the modeling.

**Contrastive Learning.** Then we can obtain the embedding of a query or document by the unified encoder. Prior works usually adopt the conventional CL loss to align the representations of queries and documents. To achieve this, we first define a function $\mathrm{sim}_\theta(\cdot, \cdot)$ to measure the similarity between two inputs $x$ and $y$ as follows,

$$\mathrm{sim}_\theta(x, y) := \frac{\mathrm{sim}[f_\theta(x), f_\theta(y)]}{\tau}, \quad (1)$$

where $\tau$ is the contrastive temperature parameter. As $f_\theta(\cdot)$ is a unified encoder, $x$ and $y$ can be in any modalities or the composition of them. Next, the CL objective (Radford et al., 2021) is optimized for aligning the query and positive document. Given the dataset $\mathcal{D}$ that is constructed by queries and documents, in which each query is paired with its positive document, the CL loss can be formalized as follows,

$$\mathcal{L}^{\mathrm{CL}} = \mathbb{E}_{x,y^+,Y \sim \mathcal{D}} \left[ -\log \frac{e^{\mathrm{sim}_\theta(x,y^+)}}{e^{\mathrm{sim}_\theta(x,y^+)} + \sum_{y \in Y} e^{\mathrm{sim}_\theta(x,y)}} \right] \quad (2)$$

where $x$ is the query, $y^+$ is the paired positive document and $Y$ is the batch of negative documents. By doing so, all examples are encoded into a shared space, within which the query is expected to be pulled closer to the positive document but pushed away from negative ones.

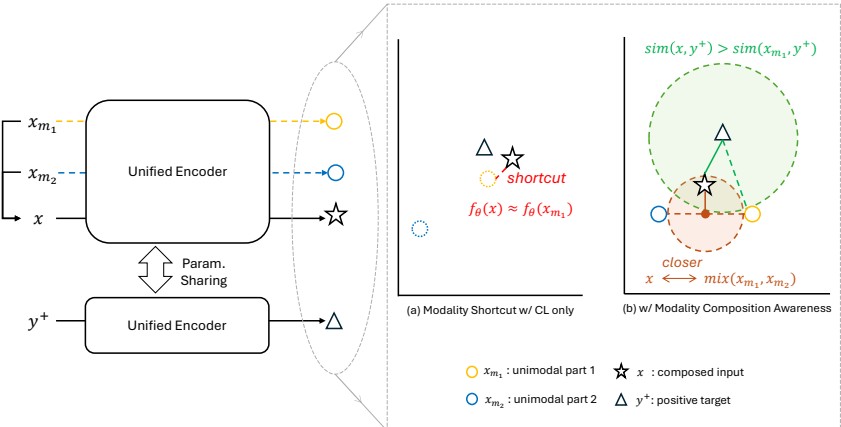

Figure 2: Modality Composition Awareness (MCA, §3.2). Unimodal parts are shown as dotted circles when not explicitly modeled. **(a)** Training with vanilla CL: the composed embedding can be close to the target but still align disproportionately with one unimodal part, leading to modality shortcuts. **(b)** Training with CL and MCA: the composed embedding is explicitly constrained to be closer to the target than any of its unimodal parts by MCP (§3.2.1), and anchored to a compositional prototype constructed from unimodal embeddings by MCR (§3.2.2) to mitigate modality shortcut.

## 3.2 MCA: MULTIMODAL EMBEDDING WITH MODALITY COMPOSITION AWARENESS

Although the conventional CL has demonstrated remarkable effectiveness in retrieval (Radford et al., 2021; Baldrati et al., 2022; Liu et al., 2023c), it is inherently designed under the separate-encoder paradigm, where each modality is encoded independently and alignment is enforced at the representation level. With the rapid development of MLLMs, the utilization of unified encoder for retrieval is becoming promising (Jiang et al., 2024; Zhang et al., 2024; Huang et al., 2025; Jiang et al., 2025). However, those MLLM-based approaches still use conventional CL as the training objective, which is less adequate under composed scenario, where queries or documents may consist of multiple modalities. Such approaches enable models to naturally handle composed inputs within a unified architecture, which could lead to modality shortcut learning as we introduced in §1. Hence, a robust multimodal retrieval paradigm, particularly one that leverages MLLMs and adapts to a wider range of composed scenarios, requires not only aligning across modalities, but also explicitly modeling the *modality composition*.

Consequently, we propose *Modality Composition Awareness* to mitigate this issue. It consists of two components: (1) *Modality Composition Preference* (MCP, §3.2.1) and (2) *Modality Composition Regularization* (MCR, §3.2.2). These two training objectives are illustrated in Figure 2 and will be introduced in the following sections. We first formalize the notation of modalities as $\mathcal{M}^1$. Given a composed input $x$, we define the set of its unimodal counterparts as follows,

$$\mathcal{U}(x) := \{x_m \mid m \in \mathcal{M}_x\}, \tag{3}$$

where $\mathcal{M}_x \subset \mathcal{M}$ is the modality set of $x$, and $x_m$ denotes its unimodal part for modality $m$. For example, if $x$ comprises text and image modalities, then $\mathcal{U}(x) = \{x_{\text{text}}, x_{\text{image}}\}$.

The modality shortcut problem can be formalized as $f_\theta(x) \approx f_\theta(x_m)$ *regardless of other modalities* after the optimization. In such a case the model only learns unimodal representation even composed inputs are provided. Formally, this key characteristic of modality shortcut is that the representation satisfies $f_\theta((x_m, x_{\text{other}})) \approx f_\theta((x_m, x'_{\text{other}}))$, meaning the model ignores variations in the other modality. As a result, when critical information resides in other modality under OOD conditions, the generalization performance could degrade. Note that modality shortcuts only arise when at least one side of the input pair is a composed input. If both the query and document are unimodal, the task reduces to standard cross-modal retrieval with minimal shortcut risk. Therefore, the following proposed losses are only applied when either the query or document is composed.

---

[1]In this paper, we use text and image as available modalities, i.e., $\mathcal{M} = \{\text{text}, \text{image}\}$, but the definition can be extended to more modalities.

### 3.2.1 MCP: MODALITY COMPOSITION PREFERENCE.

We propose a preference loss that explicitly enforces composed inputs to be more discriminative than their unimodal counterparts. Concretely, given a pair where either side is a composed input, we compute similarities between the composed embedding and candidate targets, as well as between its corresponding unimodal parts and the same target. The MCP loss is formalized as,

$$\mathcal{L}^{\text{MCP}} = \mathbb{E}_{x,y^+ \sim \mathcal{D}} \left[ - \sum_{x_m \in \mathcal{U}(x)} \log \frac{e^{\text{sim}_\theta(x,y^+)}}{e^{\text{sim}_\theta(x_m,y^+)}} \right], \tag{4}$$

where $x$ and $y^+$ are the paired query-document sampled from dataset $\mathcal{D}$. The MCP loss encourages the composed similarity higher than the unimodal similarities, ensuring that the model leverages complementary signals from multiple modalities rather than relying on one dominant modality. In other words, the loss formulates a preference-style constraint that whenever a composed input and its unimodal counterparts compete on the same retrieval task, the composed representation should be preferred. For symmetry, we compute the loss in two directions, i.e., $x$ is the document and $y^+$ is the paired query, ensuring the preference is enforced for both query and document representation.

### 3.2.2 MCR: MODALITY COMPOSITION REGULARIZATION

In addition to the preference constraint, we further propose MCR loss to encourage the consistency between the composed embedding and a simple prototype composition of its unimodal embeddings. The intuition is that the representation produced by the encoder for a multimodal input should not deviate arbitrarily from the semantic space spanned by its unimodal parts. To this end, we define a mixer module $\text{mix}(\cdot)$ that aggregates the unimodal embeddings into a composed prototype $\mathbf{h}' \in \mathbb{R}^d$ with the same output dimension as $f_\theta(\cdot)$. Then we redefine the similarity function as follows,

$$\text{sim}^{\text{mix}}_{\theta,\phi}(x, \mathcal{U}(x')) := \frac{\text{sim}[f_\theta(x), \text{mix}_\phi(\{f_\theta(x'_m) \mid x'_m \in \mathcal{U}(x')\})]}{\tau}, \tag{5}$$

where $x$ and $x'$ are two inputs to compare. $x$ is the one we want to regularize, and $\mathcal{U}(x')$ denotes the set of unimodal components that we defined in Equation 3. Thus, the redefined similarity function measures the similarities between the composed prototype derived from $x'$ and the composed embedding of $x$. In addition, to ensure that the regularization does not dominate learning, the mixer is deliberately kept simple, such as mean pooling or gated fusion, so that the encoder must learn to align composed inputs with the composed prototype rather than overfitting to the mixer.

The MCR loss is then defined as a contrastive objective, where the composed embedding is pulled closer to its mixed prototype than to other in-batch negatives,

$$\mathcal{L}^{\text{MCR}} = \mathbb{E}_{x,X \sim \mathcal{D}} \left[ - \log \frac{e^{\text{sim}^{\text{mix}}_{\theta,\phi}(x,\mathcal{U}(x))}}{e^{\text{sim}^{\text{mix}}_{\theta,\phi}(x,\mathcal{U}(x))} + \sum_{x' \in X} e^{\text{sim}^{\text{mix}}_{\theta,\phi}(x,\mathcal{U}(x'))}} \right], \tag{6}$$

where $x$ is a sampled query or document, and $X$ is a batch of composed inputs that could be either queries or documents for symmetry. Intuitively, the MCR loss enforces that the composed embedding remains anchored to the space formed by its constituent unimodal embeddings, thereby reducing the risk of spurious modality shortcut learning. Finally, we combine the CL loss with the two proposed losses. The overall training objective for optimizing $\theta$ and $\phi^2$ is formulated as,

$$\mathcal{L} = \mathcal{L}^{\text{CL}} + \alpha \times \mathcal{L}^{\text{MCP}} + \beta \times \mathcal{L}^{\text{MCR}}, \tag{7}$$

where $\alpha$ and $\beta$ are weighting coefficients controlling the relative strength of auxiliary terms. The impact of weighting is discussed in §4.2.

## 4 EXPERIMENTS

To evaluate the proposed MCA framework, we conduct extensive experiments on multimodal retrieval. The details of the datasets and benchmarks are introduced in §A.2. The default training settings are introduced in §A.3.

---

[2]$\phi$ is optimized only when the mixer contains learnable parameters.

For evaluation we consider both IND benchmarks, which share the same distribution as the training data, and OOD benchmarks that test the generalization to unseen compositions, domains and tasks. OOD performance directly evaluates the shortcut learning problem (Geirhos et al., 2020), as also shown in Figure 3, the ideal solution generalizes to all OOD test sets, but a shortcut-driven model may only performs well on training and IND test sets. The OOD benchmarks cover two types of tasks: *OOD retrieval* that measures performance under distribution shifts such as domain and modality composition changes, and *OOD grounding* measures cross-task generalization. Both types of tasks require robust modality composition. Detailed evaluation settings are introduced in §A.4

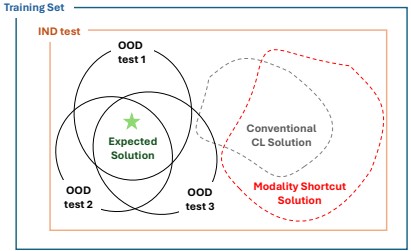

Figure 3: A conceptual diagram showing how OOD benchmarks evaluate modality shortcut.

### 4.1 MAIN RESULTS

#### 4.1.1 CONVERGENCE ANALYSIS

**Loss Curve** We first examine whether the proposed objectives affect the optimization dynamics of CL loss. Figure 4a compares the curves of the CL loss values with and without MCA during the training. Both exhibit smooth and monotonic decrease, indicating that introducing MCA does not hinder the convergence of the main contrastive objective. We then analyze the internal loss dynamics of MCA in Figure 4b. The CL, proposed MCP and MCR losses all converge stably. We note that both MCR and MCP losses exhibit noticeable fluctuations at the beginning of training. The possible reason could be unstable alignment between unimodal and multimodal embeddings in the early stage. Once the representation space becomes more coherent, both losses stabilize the remain at small magnitudes.

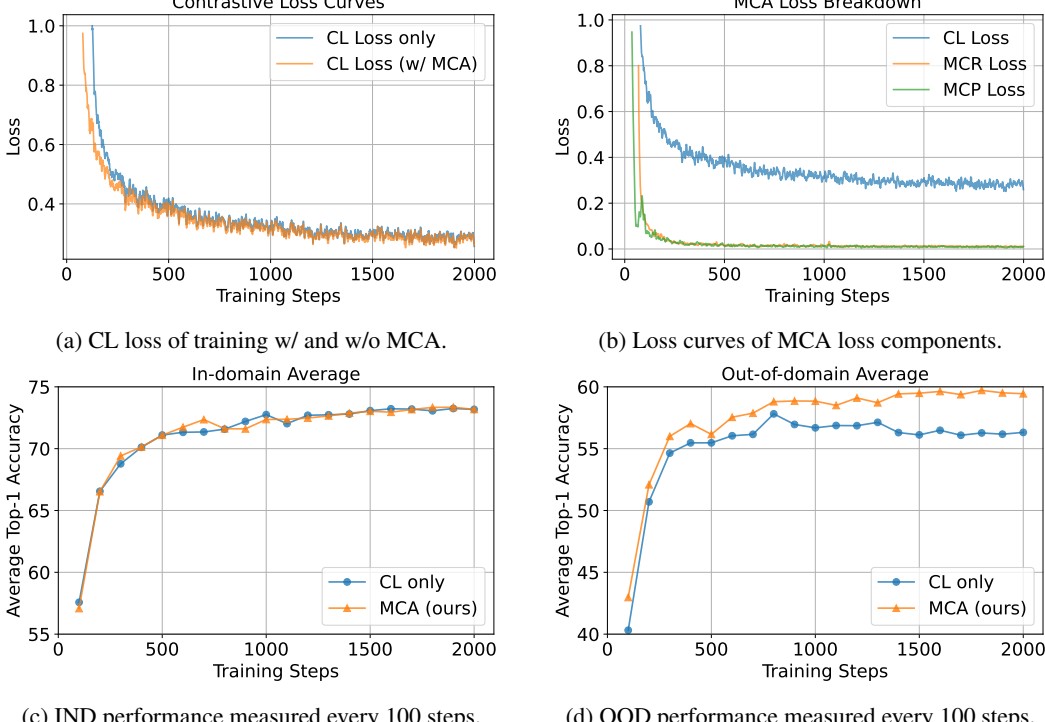

(a) CL loss of training w/ and w/o MCA.

(b) Loss curves of MCA loss components.

(c) IND performance measured every 100 steps.

(d) OOD performance measured every 100 steps.

Figure 4: Convergence analysis of MCA. For a fair comparison, the y-axis range for the loss is set to 0–1, and the range for benchmark performance is consistently set to 20.

**Performance Curve**  To further track model performance during training, we measure average retrieval accuracy every 100 training steps. Figure 4c shows the IND results, where performance improves steadily and eventually converges to a level comparable to the baseline. In contrast, Figure 4d presents the OOD results, where MCA consistently outperforms the baseline. Moreover, the performance curves reveal that the benefits of MCA appear after a few hundred of steps. While IND accuracy converges similarly for models trained with and without MCA, the ODD gap emerges within 500 steps and continues to widen throughout training. This suggests that MCA enhances generalization by mitigating modality shortcuts. Above results demonstrate that MCA converges smoothly, with auxiliary losses behaving as intended regularizer, and leads to consistent performance improvements on OOD benchmark where the robustness is most critical.

### 4.1.2 OVERALL AND BREAKDOWN PERFORMANCE

We further analyze empirical effectiveness of MCA and also the contribution of the two auxiliary losses separately. As shown in Figure 5, both MCP and MCR individually lead to consistent positive gains on OOD benchmarks. Notably, their standalone improvements are relatively smaller. When combined, the two losses yield substantially larger gains of +5.9% on OOD retrieval and 5.4% on zero-shot grounding, over the model trained with only contrastive learning, demonstrating the necessity of applying them together. This observation aligns with our design intuition in §3.2 that MCP encourages preference against unimodal shortcuts while MCR enforces structural consistency, and only their joint application forms the ideal constraint for robust modality composition, as illustrated in Figure 2. On IND benchmarks, all model variants converge to almost identical accuracy levels. This indicates that MCA acts as a lightweight regularizer. It preserves in-domain performance while delivering substantial gains under OOD conditions. The overall results contain specific numbers for each benchmark are introduced in §A.5.

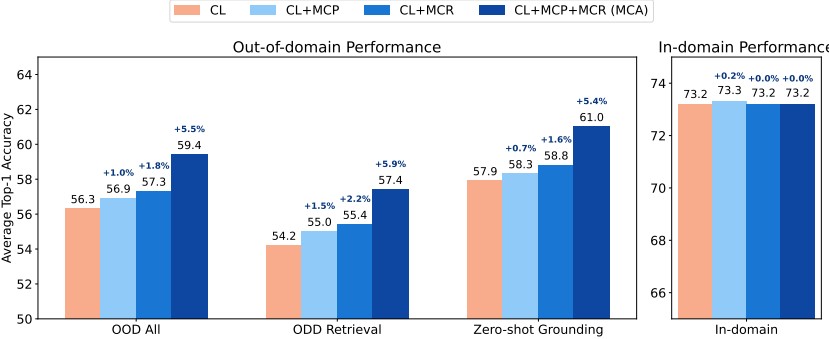

Figure 5: Breakdown of MCA though loss ablation.

### 4.2 IMPACT OF INPUT IMAGE RESOLUTION AND WEIGHTING

To better understand the behavior of MCA, we investigate its sensitivity to two critical factors that are relevant: the resolution of image inputs and the weighting of proposed losses. These two factors directly affect the balance between modalities. Specifically, when input information is degraded, the regularization strength might become crucial prevent shortcuts, while richer input information may require lighter regularization to avoid over-constraining the model. Table 1 summarizes the results across three input image resolution and different weighting values. For simplicity, we keep the coefficients of MCP and MCR loss equal, and very their shared ratio to the CL loss.

Overall, we observe that higher resolution consistently yield stronger performance as expected, since richer visual detail provides more reliable visual grounding. Interestingly, the relative gain from MCA grows larger as the resolution decreases. In lower resolution setting, where visual information is degraded and the risk of modality shortcuts is higher, MCA provides significant improvements by enforcing the use of complementary textual cues. This patter suggest that MCA is potentially more effective in scenarios with *imbalanced modality* quality, where it helps the model resist collapsing onto a single modality and instead maintain faithful composition. Nevertheless, we also observe notable exceptions. In particular, under low-resolution inputs with very small weights 0.01, the

Table 1: Impact of weighting MCA on varying input image resolution. Darker colors denote larger deviations from the vanilla CL.

| Ratio of MCA | In-domain Avg. | OOD Avg. | OOD Retrieval Avg. | Zero-shot Grounding Avg. |
|---|---|---|---|---|
| *Low resolution (128 × 128)* | | | | |
| 0 (Vanilla CL) | 69.7 | 46.7 | 42.5 | 49.9 |
| 0.01 | 69.5 (-0.3%) | 44.0 (-5.8%) | 37.6 (-11.6%) | 48.8 (-2.2%) |
| 0.10 | 69.3 (-0.6%) | 49.5 (+5.9%) | 46.8 (+10.1%) | 51.4 (+3.0%) |
| 1.00 | 69.5 (-0.3%) | 50.9 (+8.9%) | 48.1 (+13.1%) | 52.9 (+6.0%) |
| *Mid resolution (672 × 672)* | | | | |
| 0 (Vanilla CL) | 72.0 | 53.9 | 49.9 | 56.8 |
| 0.01 | 71.3 (-1.0%) | 54.7 (+1.4%) | 53.5 (+7.2%) | 55.5 (-2.3%) |
| 0.10 | 71.5 (-0.7%) | 54.9 (+1.8%) | 51.3 (+2.8%) | 57.6 (+1.4%) |
| 1.00 | 70.9 (-1.6%) | 55.8 (+3.5%) | 53.9 (+8.0%) | 57.3 (+0.8%) |
| *High resolution (1344 × 1344)* | | | | |
| 0 (Vanilla CL) | 73.2 | 56.3 | 54.2 | 57.9 |
| 0.01 | 73.2 (+0.0%) | 59.4 (+5.5%) | 57.4 (+5.9%) | 61.0 (+5.4%) |
| 0.10 | 73.0 (-0.3%) | 56.6 (+0.5%) | 53.4 (-1.5%) | 59.1 (+2.0%) |
| 1.00 | 73.2 (+0.0%) | 58.3 (+3.5%) | 57.0 (+5.1%) | 59.3 (+2.4%) |

model performs significantly worse than the vanilla CL baseline on OOD benchmarks. The possible reason is, the additional losses introduce small but inconsistent gradients, which are amplified by the noisy low-resolution image representations. As a result, the model is shifted away from the vanilla CL optimum without receiving sufficient corrective signal, causing a collapse even more severe than the baseline. In the mid-resolution setting, we also observe a modest degradation on IND benchmarks. this can be interpreted as the classic bias-variance trade-off. This degradation remains small, and the overall benefit of MCA becomes more evident when robustness is prioritized.

In addition, the results generally highlight an interaction between input richness and the strength of MCA regularization. When visual inputs are degraded, the risk of modality shortcuts is amplified, and stronger weighting is required for MCA to be effective. In contrast, when inputs provide richer information, the model relies less on shortcuts and lighter weighting suffices to stabilize the training. This trend suggests a practical guideline: the richer the input information, the smaller the weighting can lead to a better generalization, whereas weaker inputs demand stronger regularization to preserve robust modality composition.

Table 2: Impact of mixer. Darker colors denote larger deviations from the vanilla CL.

| Mixer | In-domain Avg. | OOD Avg. | OOD Retrieval Avg. | Zero-shot Grounding Avg. |
|---|---|---|---|---|
| Vanilla CL | 73.2 | 56.3 | 54.2 | 57.9 |
| Mean Pooling | 73.3 (+0.1%) | 56.5 (+0.3%) | 54.8 (+1.1%) | 57.8 (-0.2%) |
| Gated Fusion | 73.2 (+0.0%) | 59.4 (+5.5%) | 57.4 (+5.9%) | 61.0 (+5.4%) |
| MFB (Yu et al., 2017) | 73.0 (-0.2%) | 57.7 (+2.4%) | 54.6 (+0.7%) | 60.0 (+3.6%) |

## 4.3 MIXER IMPLEMENTATION

To examine the effect of different implementations of the mixer that defined in Equation 5, we compare several simple modules, including the gated fusion module by default, the non-parametric mean pooling and the multimodal factorized bilinear pooling (MFB) (Yu et al., 2017). All mixer choices introduce only a negligible number of additional parameters, ensuring that the observed gains are not due to increased model capacity. As shown in Table 2, the choice of mixer has little effect on IND benchmarks, suggesting that MCA does not alter the standard retrieval behavior. However, the improvements differ in OOD settings. Both gated fusion and MFB variants yield improvements over the CL-only baseline, demonstrating the explicitly anchoring composed embeddings to the prototype from their unimodal parts is beneficial. In contrast, the mean pooling provides only marginal improvements and even slightly degrades perfromance on cross-task grounding. This indicates that overly simple designs may be insufficient in practice to construct the prototype from unimodal embeddings. Among them, the default gated fusion achieves the best performance. Above experiments suggest that the effectiveness of MCR does not heavily rely on the specific choice of the mixer, but

rather on the general principle of enforcing compositional consistency, although the gain could vary depending on the mixer implementation.

## 4.4 QUALITATIVE ANALYSIS

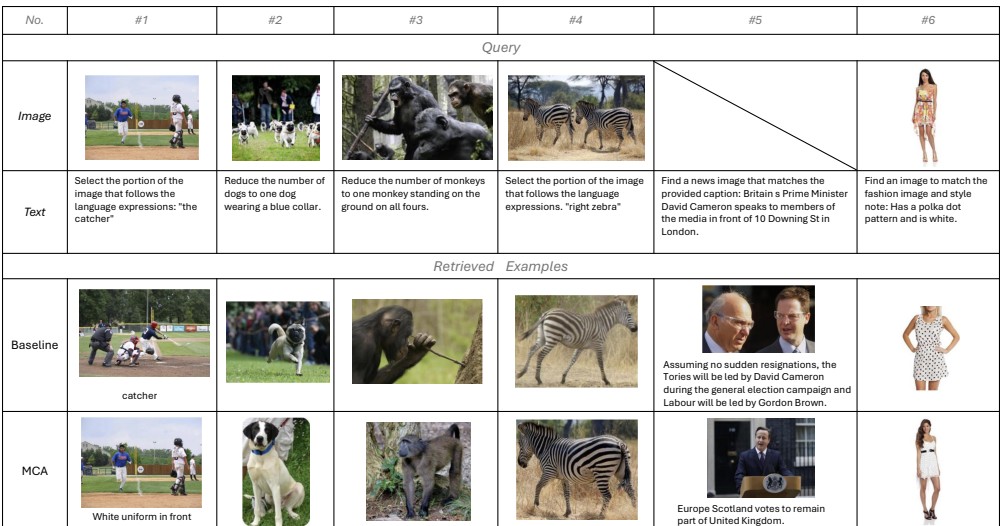

Figure 6: Qualitative examples.

Figure 6 presents qualitative comparisons between the baseline, i.e., training with only contrastive learning, and MCA model, to highlight the behavior of modality shortcut. Across diverse cases from multiple benchmarks, the baseline fails by relying on partial cues in dominant modalities. For example, in #1 it select an irrelevant baseball scene because of the word *"catcher"*, ignoring the visual grounding on the original image. In contrast, MCA retrieves the image with textual description *"white uniform in front"*. In #2 and #3, the baseline retrieves examples relying on image similarity only, but ignoring the textual instructions. *"wearing a blue collar"* in #2 and *"standing on the ground on all fours"* in #3 are not reflected, while MCA grasps those information. In #4, the baseline mistakenly retrieves a zebra facing right, over-relying on the text cue and failing to ground the original image. MCA, by contrast, localizes the *"right zebra"* in the scene. In the news retrieval case #5, the query mentions *"David Cameron speaking in front of 10 Downing St"*, yet the baseline retrieves the pair containing *"David Cameron"* in both text and image but ignores the *" speaking in front of 10 Downing St"*. MCA composes the texts and image thus finds the best match. Similarly, in the fashion example #6, the baseline retrieves an image satisfying *"polka dot and white"* but ignoring the original clothing style in the image. In contrast, MCA integrates both modalities and selects the desired target. Those examples demonstrate how MCA mitigate modality shortcuts with the potential for more robust multimodal retrieval under various composed scenarios. In addtion, we also present visualization of learned embeddings demonstrating that MCA effectively reduce the modality shortcut in §A.6.

## 5 CONCLUSION

In this work, we introduced modality composition awareness (MCA) for robust multimodal retrieval. By explicitly modeling the relationship between unimodal and multimodal inputs, MCA incorporates two objectives: (1) modality composition preference, which discourages modality shortcuts by ensuring that composed representations are more discriminative than unimodal ones, and (2) modality composition regularization , which stabilizes embedding by aligning multimodal presentations with compositional prototypes. Extensive empirical experiments across IND and OOD benchmarks demonstrate that MCA achieves consistent gains under distribution shifts and unseen zero-shot tasks, while maintaining comparable IND accuracy. These results highlight MCA as an effective principle for mitigating modality shortcut problem when using MLLMs as unified encoder and improving generalization in multimodal retrieval.

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

# A APPENDIX

## A.1 LLM USAGE

During the preparation of this paper, we utilized LLMs to assist with English proofreading. We also use LLM assistant for improving the efficiency of experiments such as organizing job execution scripts. Additionally, we used LLM assistant to support processing experimental results and generating figures.

## A.2 DATASETS

**Training data.** We utilize six multimodal retrieval datasets including both *cross-modal* and *composed* tasks for training: MSCOCO (Lin et al., 2014), VisualNews (Liu et al., 2021a), VisDial (Das et al., 2017), CIRR (Liu et al., 2021b), NIGHTS (Fu et al., 2023), and WebQA (Chang et al., 2022). Table 3 shows the number for examples and input modalities of training data. We use the training splits from MMEB[3] without additional curation. All models are trained jointly on the union of these datasets. Although MCA is designed for composed inputs, unimodal training pairs are also included. Each training batch contains a mix of unimodal and composed inputs. This ensures that the model learns basic representation ability for each modality, which is essential modeling of structural relationships among modalities that is required by our objectives.

Table 3: Statistics of training datasets. I and T are the abbreviation of image and text, respectively.

| Dataset | MSCOCO | | VisualNews | | VisDial | CIRR | NIGHTS | WebQA |
|---|---|---|---|---|---|---|---|---|
| Input modalities | $I \rightarrow T$ | $T \rightarrow I$ | $I \rightarrow T$ | $T \rightarrow I$ | $T \rightarrow I$ | $(I+T) \rightarrow I$ | $I \rightarrow I$ | $T \rightarrow (I+T)$ |
| # of training pairs | 113K | 100K | 100K | 100K | 123K | 26K | 16K | 17K |

**Benchmarks.** We evaluate the models on a range of multimodal retrieval benchmarks, which we group into *IND* and *out-of-domain* (OOD) settings. The IND benchmarks correspond directly to the test splits of the six datasets used for training as mentioned in the previous paragraph and Table 3. Since models that rely on modality shortcuts often exhibit poor generalization, we also assess this issue through robustness under OOD settings. The OOD benchmarks are deliberately selected: OVEN (Hu et al., 2023) combines visual and textual modalities, where each instance consists of an image paired with a visual recognition question, alongside a reference Wikipedia image and its textual description including the title and first 100 tokens that serve as the target candidate for answering the question. FashionIQ (Hu et al., 2023) features a multimodal composition of fashion product images paired with crowd-sourced textual descriptions that specify differences between products, enabling composed image retrieval where a reference image and modification text are combined to retrieve target images. EDIS (Liu et al., 2023b) combines entity-rich text queries with multimodal candidates consisting of news images paired with their headlines, requiring models to understand both textual entities/events and visual content for retrieval. MSCOCO-Grounding (Lin et al., 2014; Jiang et al., 2025) transforms object detection into a multimodal ranking task where queries combine an image with a textual object name to retrieve cropped images of the specified object, with distractors sourced from other objects within the same image and from different images. Visual7W-Pointing (Zhu et al., 2016) combines textual questions with images to establish semantic links between descriptions and image regions, enabling both visual question answering through text-image composition and visual grounding through multimodal object localization. RefCOCO (Kazemzadeh et al., 2014) employs multimodal queries combining images with referring language expressions to identify specific objects, pairing image-text queries to retrieve cropped object images. RefCOCO-Matching (Kazemzadeh et al., 2014; Jiang et al., 2025) uses the same source

---

[3] https://huggingface.co/datasets/TIGER-Lab/MMEB-train

datasets, while repurposed to match identical image-text compositions where both query and target contain the same object with its referring expression. We directly use the test splits from MMEB[4]. Note that since modality shortcut in MLLMs, the problem we study in this paper, only happens in composed queries or documents, all of our OOD benchmarks focus on *composed retrieval*, in which either query or document side comprise multiple modalities.

## A.3 DEFAULT TRAINING SETTINGS

By default, we adopt `Qwen2-VL-2B-Insturct`[5] as the unified encoder backbone. The contrastive temperature $\tau$ is set to 0.02. All models are fine-tuned with LoRA (Hu et al., 2022) adapter. Unless otherwise specified, the LoRA rank is set to 8. The overall loss combines conventional contrastive learning and our proposed MCR and MCP terms as introduced in Equation 7, where $\alpha$ and $\beta$ are weighting hyper-parameters. These auxiliary terms act as regularization signals rather than main training objectives. We empirically adopt a small coefficient between 0.01 and 1.0 to stabilize training. For simplicity, unless the otherwise specified, we tie $\alpha$ and $\beta$ by setting $\alpha = \beta = 0.01$ by default so that both the preference and regularization losses contribute equally. Following previous work (Jiang et al., 2025), we use AdamW optimizer with a learning rate of $2e-5$ and a linear schedule with warm-up. We also fixed the global batch size as 1024 with gradient accumulation for varying numbers of GPU and memory. The total training steps are 2000 and warm-up steps are 200. The training takes around 200 hours on $8\times$ Nvidia A100 40G. For MCR loss, the mixer function $\text{Mix}_\phi$ in Equation 5 is instantiated as a simple gated fusion module by default, where each unimodal embedding is reweighted by a learnable gating coefficient before aggregation. For ablation, we also experiment with alternatives such as non-parametric mean pooling and multimodal factorized bilinear pooling (Yu et al., 2017) in §4.3.

## A.4 DEFAULT EVALUATION SETTINGS

Unless otherwise specified, we report results using the final checkpoint of training for a fair comparison across all methods, as we have verified that the trend is stable across checkpoints in §4.1.1. We report standard retrieval metrics accuracy@1 for all tasks.

## A.5 OVERALL RESULTS

Table 4: Overall results

| | In-domain Retrieval | | | | | | | | | OOD Retrieval | | | | Zero-shot Grounding | | | | |
|---|---|---|---|---|---|---|---|---|---|---|---|---|---|---|---|---|---|---|
| Method | VisDial | CIRR | VisualNews.t2i | VisualNews.i2t | MSCOCO.t2i | MSCOCO.i2t | NIGHTS | WebQA | Avg. | OVEN | FashionIQ | EDIS | Avg. | MSCOCO | Visual7W-P | RefCOCO | RefCOCO-M | Avg. |
| *Dual-encoder Approaches (not trained in the same setting, just for reference)* | | | | | | | | | | | | | | | | | | |
| CLIP | 30.7 | 12.6 | 78.9 | 79.6 | 59.5 | 57.7 | 60.4 | 67.5 | 55.8 | 41.1 | 11.4 | 81.0 | 44.5 | 33.8 | 55.1 | 56.9 | 61.3 | 51.7 |
| SigLIP | 21.5 | 15.1 | 51.0 | 52.4 | 58.3 | 55.0 | 62.9 | 58.1 | 46.7 | 56.0 | 20.1 | 23.6 | 46.4 | 33.2 | 70.8 | 50.8 | 70.1 | 56.2 |
| BLIP2 | 18.0 | 9.8 | 48.1 | 13.5 | 53.7 | 20.3 | 56.5 | 55.4 | 39.5 | 39.3 | 9.3 | 54.4 | 34.4 | 28.9 | 47.4 | 59.5 | 52.0 | 46.9 |
| *Comparable Unified-encoder Approaches* | | | | | | | | | | | | | | | | | | |
| *Low resolution (128 × 128)* | | | | | | | | | | | | | | | | | | |
| Vanilla CL | 72.7 | 50.2 | 73.0 | 74.2 | 68.1 | 67.5 | 65.7 | 86.5 | 69.7 | 45.4 | 15.4 | 66.6 | 42.5 | 35.5 | 45.3 | 52.3 | 66.4 | 49.9 |
| CL + 0.01 × MCA | 74.8 | 49.9 | 71.3 | 75.3 | 70.3 | 66.1 | 63.3 | 85.1 | 69.5 | 43.1 | 14.4 | 55.4 | 37.6 | 36.0 | 45.1 | 46.2 | 68.0 | 48.8 |
| CL + 0.10 × MCA | 75.1 | 46.2 | 72.0 | 74.8 | 69.3 | 66.9 | 64.6 | 85.1 | 69.3 | 55.0 | 12.3 | 73.2 | 46.8 | 37.8 | 47.4 | 52.4 | 68.1 | 51.4 |
| CL + 1.00 × MCA | 74.0 | 47.2 | 71.6 | 75.0 | 67.3 | 67.7 | 65.5 | 87.7 | 69.5 | 57.8 | 14.1 | 72.4 | 48.1 | 39.2 | 51.7 | 51.9 | 68.9 | 52.9 |
| *Mid resolution (672 × 672)* | | | | | | | | | | | | | | | | | | |
| Vanilla CL | 78.5 | 50.7 | 74.2 | 77.0 | 71.3 | 71.1 | 66.5 | 86.9 | 72.0 | 58.9 | 17.2 | 73.6 | 49.9 | 41.8 | 57.8 | 57.4 | 70.3 | 56.8 |
| CL + 0.01 × MCA | 79.9 | 47.4 | 74.4 | 76.6 | 71.1 | 70.0 | 63.5 | 87.2 | 71.3 | 61.7 | 12.3 | 86.6 | 53.5 | 40.3 | 56.2 | 59.3 | 66.3 | 55.5 |
| CL + 0.10 × MCA | 77.7 | 49.5 | 73.8 | 77.1 | 71.9 | 69.1 | 64.8 | 87.9 | 71.5 | 62.1 | 14.3 | 77.4 | 51.3 | 40.5 | 58.0 | 63.4 | 68.6 | 57.6 |
| CL + 1.00 × MCA | 78.0 | 45.5 | 74.1 | 76.5 | 68.6 | 71.1 | 66.3 | 87.4 | 70.9 | 64.0 | 16.0 | 81.7 | 53.9 | 39.8 | 59.7 | 62.0 | 67.7 | 57.3 |
| *High resolution (1344 × 1344)* | | | | | | | | | | | | | | | | | | |
| Vanilla CL | 81.3 | 51.6 | 75.7 | 78.6 | 73.9 | 71.9 | 65.6 | 86.8 | 73.2 | 63.0 | 15.6 | 84.0 | 54.2 | 41.0 | 60.7 | 60.3 | 69.6 | 57.9 |
| CL + 0.01 × MCA | 80.3 | 50.5 | 76.3 | 78.7 | 74.6 | 72.1 | 66.1 | 86.8 | 73.2 | 66.7 | 19.3 | 86.1 | 57.4 | 42.7 | 65.4 | 65.0 | 70.9 | 61.0 |
| CL + 0.10 × MCA | 81.3 | 51.0 | 74.6 | 78.9 | 73.5 | 71.2 | 66.0 | 87.7 | 73.0 | 62.4 | 16.3 | 81.6 | 53.4 | 41.7 | 63.4 | 63.3 | 67.8 | 59.0 |
| CL + 1.00 × MCA | 81.4 | 49.1 | 75.6 | 78.4 | 73.9 | 73.1 | 66.8 | 87.3 | 73.2 | 66.2 | 18.7 | 86.1 | 57.0 | 40.1 | 62.8 | 63.5 | 70.8 | 59.3 |

---

[4] https://huggingface.co/datasets/TIGER-Lab/MMEB-eval

[5] https://huggingface.co/Qwen/Qwen2-VL-2B-Instruct

For completeness, we report the detailed performance of MCA under all weighting and resolution settings across each benchmark. Table 4 provides the full numbers for IND retrieval as well as OOD benchmarks, including both OOD retrieval and zero-shot grounding. These results complement the aggregated scores in §4 with the trends discussed: (1) the richer the input information, the smaller the weighting can lead to a better generalization, whereas weaker inputs demand stronger regularization to preserve robust modality composition; (2) higher resolutions consistently yield stronger baselines; (3) IND degradations remain small across all settings.

## A.6 TSNE IMPLEMENTATION

In Figure 1b and 1c, we visualize the learned embedding spaces using t-distributed Stochastic Neighbor Embedding (t-SNE) to analyze the distribution characteristics of different query types. Embeddings were extracted from the CIRR dataset, which contains image-text compositions for retrieval tasks. For dimensionality reduction, we employed t-SNE with perplexity=100, 1,000 iterations, and a random seed of 42 to ensure reproducibility. We randomly sampled 300 instances from each embedding type to maintain visual clarity while preserving distribution characteristics. To ensure fair comparison, we used identical sampling indices across all embedding types for baseline and our model. The visualization combines kernel density estimation (KDE) with scatter plots to represent both the overall distribution and individual data points. We implemented a custom transparency gradient for the KDE plots, making low-density regions increasingly transparent to highlight areas of embedding concentration. Different marker shapes distinguish between embedding types: circles for composed queries, squares for text-only queries, triangles for image-only queries, and diamonds for target images. The resulting visualization effectively captures the relationships between different modalities in the embedding space, revealing patterns of overlap and separation between composed, unimodal, and target representations. Comparing Figure 1b and 1c, it clearly indicates that vanilla CL could lead to a over-mixed distribution for composed and text-only embeddings, while MCA learns a more robust representation with a clear boundary among different types of representations with a competitive IND performance and a better OOD performance.

## A.7 LIMITATION AND FUTURE DIRECTIONS

While our proposed modality composition awareness show consistent improvements across various OOD benchmarks, several limitations remain. First, the design of our losses assume that modality shortcuts primarily arise when composed inputs are present, and thus the framework is not directly applied to purely unimodal queries. This assumption may overlook shortcut phenomena that could still exist in separate-encoder approaches, although the risk is notably lower. Second, the experimental results still show that our proposed MCA could lead to performance degradation under particular weighting and input resolution setting. Although rare, this phenomenon suggests that it may requires setting adjust to make the proposed loss work in practice. Third, in this work we only explore simple mixer implementation for fair comparison, and extending the design to richer mixers is an interesting open direction. Lastly, although we have identified and studied the modality shortcut problem for vision-text, due to the lack of composed training data and benchmarks, the generalization to broader modalities, e.g., audio, has not yet been validated. We expect the problem of modality shortcut to become even more prominent in the future with more available composed training data and benchmarks in MLLM-based retrieval.

## A.8 ETHICAL STATEMENT

This work focuses on improving multimodal representation learning by mitigating modality shortcut problem. Our experiments are conducted entirely on publicly available datasets and model checkpoints. Our method does not directly generate new contents but instead focuses on retrieval, as a result, it poses minimal risk of misuse in generating harmful or deceptive contents. Nevertheless, we emphasize the importance of responsible use of the retrieval model.

