# OpenReview forum: "MCA: Modality Composition Awareness for Robust Composed Multimodal Retrieval"
_ICLR.cc/2026/Conference — ICLR 2026 Conference Withdrawn Submission_

### Official Review · Reviewer_jWKA · 2025-10-29

**Soundness:** 2
**Presentation:** 3
**Contribution:** 2
**Rating:** 4
**Confidence:** 5

**Summary:**

The paper proposes Modality Composition Awareness, a training framework to mitigate the “modality shortcut” problem when using Multimodal Large Language Models as a unified encoder for composed multimodal retrieval. MCA has two complementary objectives: 1.Modality Composition Preference, a preference-style loss that enforces composed embeddings to be more discriminative than any unimodal counterpart. 2. Modality Composition Regularization, aligns the composed embedding with a prototype assembled from unimodal embeddings via a simple mixer and a contrastive objective.  Extensive experiments on IND and OOD benchmarks show MCA preserves IND accuracy while improving OOD robustness. The paper also studies sensitivity to image resolution, loss weighting, and mixer variants.

**Strengths:**

1.	The paper formalizes the “modality shortcut” problem for unified MLLM encoders under composed inputs
2.	MCP (preference) and MCR (consistency to mixed prototype) are conceptually simple and can be integrated as light-weight regularizers to contrastive training

**Weaknesses:**

1.	It seems that an additional empirical experiment is needed to directly verify whether the proposed method effectively mitigates the modality shortcut problem.
2.	The paper lacks sufficient discussion from the textual perspective (i.e., how the method affects or interacts with text representations).
3.	The paper lacks a relevant benchmark to empirically validate the problem mentioned by the authors, eg modality shortcut.
4.	The paper is not sufficiently comprehensive; additional text-related experiments or analyses of the model’s training and testing behaviors would make the work more complete.
5.	The paper lacks comparisons with existing open-source models. Since the evaluation is conducted only against models trained by the authors themselves, it is difficult to verify the effectiveness of the proposed method and to confirm whether the targeted research problem truly exists.

**Questions:**

1.	Please quantify additional time/memory cost of MCP/MCR during training compared to vanilla contrastive learning.
2.	Could you provide more qualitative examples and visualizations (score histograms, embedding distance changes) in the appendix to illustrate how MCA modifies representations?
3.	It seems that an additional empirical experiment is needed to directly verify whether the proposed method effectively mitigates the modality shortcut problem. Moreover, the paper lacks sufficient discussion from the textual perspective (i.e., how the method affects or interacts with text representations).
4.	Is the modality shortcut problem empirically verified to exist in the current datasets? If it indeed exists, why does the proposed method improve performance only on out-of-domain benchmarks but not on in-domain ones? It would be more convincing if the authors could collect or annotate a dedicated benchmark that explicitly reflects this issue.
5.	Do existing open-source models also exhibit the modality shortcut problem? Why not include comparisons with these models to empirically demonstrate that the issue truly exists?

---

### Official Review · Reviewer_ny7F · 2025-10-31

**Soundness:** 1
**Presentation:** 3
**Contribution:** 2
**Rating:** 2
**Confidence:** 3

**Summary:**

The paper proposes the MCA framework to address the modality shortcut problem present in composed retrieval scenarios within current MLLMs. This framework introduces two auxiliary losses: the Modality Composition Preference (MCP) loss and the Modality Composition Regularization (MCR) loss. Through experiments, the paper demonstrates that MCA can improve the effectiveness of out-of-distribution retrieval while maintaining in-domain performance.

**Strengths:**

1.The paper provides a explanation of the modality shortcut problem in current MLLMs. By using conceptual examples and t-SNE visualizations, it shows that this problem exists in large models and allows readers to understand the definition of the problem well.

2.The paper is well-written and lists detailed experimental settings, which facilitates reproducibility.

3.The paper's experiments are comprehensive, and the effectiveness of the method is verified through experimental analysis.

**Weaknesses:**

1.The method's performance on IND tasks is not outstanding. Although MCA performs well in OOD scenarios, it can be seen from Table 1 that the method shows a consistent decline in IND scenarios, indicating that the method is not sufficiently robust.

2.The discussion of modalities is limited in scope. Although the paper mentions using text and image modalities as representatives, other modalities have different characteristics, and the conclusions drawn in the paper may not be representative. Scenarios involving three or more modalities might also be more complex, and the paper lacks discussion on this as well.

3.The figures in the paper are not clear enough. Figures 2 and 3 have large areas of white space, while the font used for illustration is too small, making the proportions uncoordinated and difficult for the reader to see clearly.

**Questions:**

1.There is a severe failure case in Table 1, where the OOD Avg performance drops by 11.6% under low resolution and low weight. Although the paper provides an explanation, I believe this change might be beyond the scope of that explanation. Could there be an error in the experimental setup? Otherwise, this sensitivity might render the method not universally applicable.

2.The MLLM used in the paper is Qwen-VL-2B-Instruct. Could this model have a relatively small number of parameters, leading to poorer multimodal alignment capabilities? If a model with a larger number of parameters were used, could this problem be directly avoided, thereby making this method lose its intended significance?

3.In Table 2, the "Gated Fusion" mixer performs the best. Could the authors please provide the specific architecture or mathematical formula for this module? It is mentioned in Appendix A.3, but its implementation details do not seem to be provided. Also in Table 2, "Mean Pooling," as the simplest combination method, shows very limited or even negative improvement in OOD performance. Does this imply that simple vector averaging is insufficient for constructing a meaningful compositional prototype?

---

### Official Review · Reviewer_o4QY · 2025-11-02

**Soundness:** 2
**Presentation:** 2
**Contribution:** 2
**Rating:** 2
**Confidence:** 3

**Summary:**

This paper studies multimodal retrieval with MLLM unified encoders and identifies a modality shortcut issue, where models over-rely on one modality instead of jointly leveraging both, leading to reduced robustness under distribution shifts. To mitigate this, the authors introduce a modality composition awareness framework, combining a preference loss to favor multimodal over unimodal representations and a composition regularization to align multimodal features with prototypes composed from unimodal parts. Experiments on several OOD retrieval benchmarks show consistent gains. Overall, the paper addresses a relevant problem with a simple and intuitive solution, but the experimental scope is limited, particularly in baseline selection, dataset diversity, and clarity of the shortcut analysis.

**Strengths:**

1. Addresses modality shortcut behavior in unified-encoder MLLMs, a relevant issue as unified retrieval becomes more common.

2. The proposed preference and composition regularization losses are easy to implement and computationally lightweight.

3. Shows consistent OOD retrieval improvements and basic ablations confirming each loss contributes.

**Weaknesses:**

1. The concept is compelling but currently presented somewhat vaguely. The illustrative figure does not intuitively convey shortcut behavior; more concrete failure case analysis would strengthen the argument.

2. Baselines are relatively old (mostly CLIP-style models). Missing comparisons with recent multimodal LLMs and retrieval-tuning frameworks would improve credibility.

3. Manual selection without principled justification or tuning range exploration weakens the training claim. Lack of theoretical or empirical guidance.

4. Although improvements are shown, the number of OOD scenarios and datasets is limited, weakening the robustness claim.

5. Strengthening the narrative, why unified encoders for retrieval matter long-term, would increase appeal.

**Questions:**

1. Can you provide a clearer formal definition and more qualitative evidence (e.g., retrieval examples showing failure due to shortcut)? Could you compare the phenomenon with known shortcut learning literature?

2. What is the sensitivity to α and β? Could you provide empirical results or theoretical reasoning for their chosen values?

3. Why not compare against stronger contemporary retrieval backbones (e.g., BLIP-2/Flamingo-style encoders, EVA-CLIP, SigLip)? Do your findings still hold when the base MLLM has already undergone multimodal alignment training?

4. Can the method apply to tasks beyond retrieval (e.g., VQA, caption-grounded generation)? Does the gain still exist on in-distribution benchmarks?

5. Could you show ablation on unimodal vs multimodal dominance (e.g., mask image vs mask text cases)?

---

### Official Review · Reviewer_YQUt · 2025-11-09

**Soundness:** 2
**Presentation:** 2
**Contribution:** 2
**Rating:** 2
**Confidence:** 4

**Summary:**

The authors develop a method, Modality Composition Awareness (MCA), for mitigating the modality shortcut problem often incurred when primarily relying on contrastive learning based methods for late-fusion and MLLM-adaptation approaches to universal/cross-modal multimodal retrieval. Specifically, to encourage having the joint model learn a robust composed representation (and avoid having a single dominant modality), the authors propose a preference loss that: (1) enforces the embeddings of a multimodal composition to be more discriminative that any of its unimodal counterparts (lines 83-84 -- "Modality Composition Preference" (MCP)); and (2) a composition regularization objective rewards consistency between the composite embedding from the unified encoder and a composite prototype constructed from the unimodal embeddings ("Modality Composition Regularization" (MCR)). Methodologically, MCP is performed by adding a loss function similar to 'standard' contrastive loss that pulls the composed embedding closer to the paired item than the unimodal similarities (Equation 4) and MCR is performed by another contrastive loss objective where the composed embedding is pulled closer to its mixed prototype (gated fusion works better than mean pooling and multimodal factorized bilinear pooling (MFB) in Table 2) than other in-batch negatives. These three losses (i.e., 'standard' CL, MCP, MCR) are added together as a linear function. Experiments are conducted on the {cross-modal, composed} task parts of MMEB, focusing on the distinction between in-domain and out-of-domain (OOD) performance differences where the OVEN, FashionIQ, EDIS are the OOD datasets and additional datasets {MSCOCO-Grounding, Visual7W-Pointing, RefCOCO, RefCOCO-Matching} are used for zero-shot grounding settings. The core empirical conclusion is that has minimal effect on the in-domain cases, but demonstrates improvements on OOD and ZS-Grounding settings (Table 4) trained used a Qwen-VL-2B-Instruct model as a backbone and reporting results as accuracy@1.

**Strengths:**

Strengths of this work include:
- The core idea is conceptually appealing for mitigating the modality shortcut problem in multimodal retrieval.
- The empirical performance on the performed experiments consistently demonstrate that there is minimal loss for in-domain settings and improvements in OOD and ZS-Grounding settings across multiple datasets.
- The 'imbalanced' modality quality experiments and associated observations are interesting from the perspective of modality shortcut issues.
- The motivation/introduction section (Section 1 + Figure 1, Figure 2) and methods section (Section 3) is well-written and easy to understand.

**Weaknesses:**

Weaknesses of this work include:
- There is far too much of the experimental details in the Appendices; the paper basically cannot be evaluated without reading the appendices. While I understand there are space constraints, having the main results in the Appendices also make the analysis and interpretation difficult to follow and validate the main claims. If I need to choose (even though I actually think the main paper can make room for both), section 4.1.1 and section 4.3 are less important than Appendix A5 and parts of A2, A3 & A4.
- In Table 4, the only baseline models provided are dual-encoder approaches and 'standard' contrastive learning. However, MCA is a MLLM-adaptation method (lines 140-143). Thus, I think comparisons are needed to other MLLM-adaptation approaches (unless there is a good reason that I am missing). For cases where other systems perform better/worse, there should be some discussion to validate that MCA is better than all of the other loss functions that researchers augment 'vanilla CL' with.
- It is claimed that this method applies beyond image-text, which is true mathematically, but not validated when it isn't that difficult to do so with MMEB-2 (and more parts of MMEB). I suppose the OOD/ZS issues would have to be considered, but this would be valuable.
- Some guidance regarding setting the hyperparameters would be useful as there is performance variance across the loss function scaling factors in Table 4.

**Questions:**

Associated with the weakness, questions include:
- How does this performance compare to other MLLM adaptation methods for cross-modal retrieval? Would you be able to hypothesize/explain any differences -- in particular within the context of modality shortcuts?
- How does MCA perform on other modality pairs?
- Any discussion regarding hyperparameters would be useful (the discussion in A.3 is limited in my opinion).

---

### Note · Authors · 2025-11-21

I have read and agree with the venue's withdrawal policy on behalf of myself and my co-authors.